# Licochalcone A Inhibits Prostaglandin E_2_ by Targeting the MAPK Pathway in LPS Activated Primary Microglia

**DOI:** 10.3390/molecules28041927

**Published:** 2023-02-17

**Authors:** Harsharan Singh Bhatia, Matthias Apweiler, Lu Sun, Julian Baron, Ashwini Tirkey, Bernd L. Fiebich

**Affiliations:** 1Neuroimmunology and Neurochemistry Research Group, Department of Psychiatry and Psychotherapy, Medical Center-University of Freiburg, Faculty of Medicine, University of Freiburg, D-79104 Freiburg, Germany; 2Institute for Tissue Engineering and Regenerative Medicine (iTERM), Helmholtz Zentrum München, 85764 Neuherberg, Germany; 3Institute for Stroke and Dementia Research, Klinikum der Universität München, Ludwig Maximilian University of Munich (LMU), 81377 Munich, Germany

**Keywords:** Licochalcone A, flavonoids, Glycyrrhiza glabra, neuroinflammation, PGE_2_, arachidonic acid, COX-2, oxidative stress, 8-iso-PGF_2α_

## Abstract

Neuroinflammation and oxidative stress are conditions leading to neurological and neuropsychiatric disorders. Natural compounds exerting anti-inflammatory and anti-oxidative effects, such as Licochalcone A, a bioactive flavonoid present in a traditional Chinese herb (licorice), might be beneficial for the treatment of those disorders. Therefore, this study aimed to investigate the anti-inflammatory and anti-oxidative effects of Licochalcone A in LPS-activated primary rat microglia. Licochalcone A dose-dependently prevented LPS-induced PGE_2_ release by inhibiting the arachidonic acid (AA)/cylcooxygenase (COX) pathway decreasing phospholipase A2, COX-1, and COX-2 protein levels. Furthermore, LPS-induced levels of the cytokines IL-6 and TNFα were reduced by Licochalcone A, which also inhibited the phosphorylation and, thus, activation of the mitogen-activated protein kinases (MAPK) p38 MAPK and Erk 1/2. With the reduction of 8-iso-PGF_2α_, a sensitive marker for oxidative stress, anti-oxidative effects of Licochalcone A were demonstrated. Our data demonstrate that Licochalcone A can affect microglial activation by interfering in important inflammatory pathways. These in vitro findings further demonstrate the potential value of Licochalcone A as a therapeutic option for the prevention of microglial dysfunction related to neuroinflammatory diseases. Future research should continue to investigate the effects of Licochalcone A in different disease models with a focus on its anti-oxidative and anti-neuroinflammatory properties.

## 1. Introduction

Neuroinflammation plays a fundamental role in the progression of different neurological and neuropsychiatric disorders, such as Alzheimer’s disease (AD), Parkinson’s disease (PD), multiple sclerosis (MS), and depression [1,2,3]. It is characterized by activated glial cells, especially microglia [4]. Microglia, originating from the yolk sac and settling in the CNS via the circulatory system [5], are resident immune cells of the CNS. They are crucial for physiological immune responses and inflammatory processes [6] but can support chronic neuroinflammation leading to neurodegenerative diseases as well [4]. They contribute to CNS development and maintain the brain’s health, homeostasis, and function in the healthy CNS [7]. Interestingly, microglia can be rapidly activated during brain trauma or brain disorders, inducing neuroinflammation. Their activation results in the upregulation of pro-inflammatory cytokines, reactive oxygen species (ROS), and derivatives of arachidonic acid (AA) modulated by multiple signaling cascades [8,9,10]. Therefore, the suppression of microglia-mediated neuroinflammation is suggested to be a promising strategy in the therapy of neurodegenerative disorders [11].

The flavonoid Licochalcone A is a bioactive component present in the root and stem of the traditional Chinese herb licorice (Glycyrrhiza glabra; Glycyrrhiza inflata) and has been described to exert anti-inflammatory effects [12,13]. Due to its anti-inflammatory and neuroprotective effects, licorice—and therefore Licochalcone A—is used to treat a variety of diseases, such as asthma, gastric ulcers, and infectious diseases in traditional medicine [13,14]. In a recent study, Licochalcone A protected the blood–milk barrier’s integrity by inhibiting mitogen-activated protein kinases (MAPKs) and the Akt/nuclear factor-kappa B (NF-κB) pathways in vivo and in vitro [15], two pathways associated with modulation of inflammation [16]. In addition, Licochalcone A showed a reduction of IL-6, IL-1β, and TNFα as well as anti-oxidative effects in LPS-treated RAW 264.7 cells and an LPS-induced mice model of acute lung injury [17]. Furthermore, Licochalcone A ameliorated behavioral dysfunction in a murine LPS-induced PD model by the inhibition of microglial activation and the prevention of dopaminergic neuron degeneration [18]. In middle-aged C57BL/6 mice, Licochalcone A enhanced T- and B-cell proliferation via the IL-17 signal pathway, improving cognitive functions [19]. In LPS-stimulated BV-2 cells, Licochalcone A inhibited prostaglandin (PG) E_2_ production via cyclooxygenase (COX)-2 dependent mechanisms [18].

PGE_2_ is generated out of AA by the enzymatic activity of cyclooxygenases and PGE synthases (PGESs). There exist two forms of cyclooxygenases, the constitutively expressed COX-1 and the inducible form COX-2 [20]. COX-1 is found in many different tissues, such as the brain, lungs, and kidneys, and produces prostanoids [21,22]. COX-2 is induced by pathogen-/damage-associated molecular patterns (PAMPs/DAMPs), such as the bacterial cell wall component lipopolysaccharide (LPS) or pro-inflammatory cytokines [23,24]. Both form PGH_2_ out of AA [25] after AA is provided by the enzymatic activity of cytosolic phospholipase A2 (cPLA2) [26]. Three types of PGES are characterized, namely the microsomal PGESs (mPGES)-1 and -2 as well as the cytosolic PGES (cPGES) [27]. mPGES-1 is overexpressed in activated microglia [28] and forms PGE_2_ out of PGH_2_ [25]. PGE_2_, a pro-inflammatory cytokine, plays a crucial role in different pathological conditions, such as AD [29], depression [30], schizophrenia [31], and other neurological and psychiatric diseases [32,33]. 

A variety of intracellular signaling pathways, such as MAPKs and NF-κB, are involved in the regulation of COX-2 and mPGES-1. As a family of serine/threonine protein kinases, MAPKs play an important role in mediating inflammation. MAPKs are, among other stimuli, activated by Toll-like receptors (TLRs) and can be differentiated in the extracellular signal-regulated kinase (ERK 1/2), p38 MAPK, and Jun N-terminal kinase (SAPK/JNK) [34]. All those kinases take part in the regulation of inflammatory responses, such as the production of cytokines, as well as cell proliferation and apoptosis via different transcription factors [16]. Especially, p38 MAPK and NF-κB seem to be associated with the regulation of the AA/COX-2/PGE_2_ cascade [35].

8-iso-PGF_2α_ derives from AA as well but is mainly generated without enzymatic catalyzation through ROS leading to lipid peroxidation, and can be used as a sensitive marker for oxidative stress [36,37]. ROS as well as reactive nitric species (RNS), such as nitric oxide (NO), can lead to neurodegeneration as well and are closely associated with inflammation with both conditions promoting each other [38]. Oxidative stress is, therefore, discussed as another pathomechanism in neurological and neuropsychiatric disorders, and targeting oxidative pathways might offer new treatment opportunities [8,39,40].

Especially chronic low-grade inflammation seems to be associated with different neuropsychiatric disorders [41]. However, previous studies used quite high LPS doses for the stimulation of cells compared to our previous works [17,18]. The current study, therefore, investigates the anti-inflammatory and anti-oxidative effects of Licochalcone A in low-dose LPS-stimulated primary rat microglia with a focus on the AA/COX-2/PGE_2_ pathway. 

## 2. Results

### 2.1. Cytotoxic Effects of Licochalcone A in Primary Rat Microglia

To exclude the cytotoxic effects of Licochalcone A, a CellTiter-Glo^®^ Luminescent Cell Viability Assay was performed. The cell viability or metabolism of primary microglia cells was significantly enhanced by stimulation with 10 ng/mL LPS (Figure 1). As expected, 10% DMSO exerted significant cytotoxic effects, reducing cell viability to around 40% of untreated cells. Licochalcone A significantly increased cell viability/metabolism in all tested concentrations, with a slight concentration-dependent effect. Therefore, all tested concentrations were used for the following experiments. 

### 2.2. Effects of Licochalcone A on PGE_2_, TNFα, and IL-6 Release

Next, we studied the effects of Licochalcone A on different inflammatory markers using EIA and ELISA. LPS strongly induced PGE_2_ release in primary rat microglia, while untreated cells and 2.5 µM Licochalcone A without LPS showed no impact on PGE_2_ synthesis (Figure 2A). Licochalcone A significantly inhibited LPS induced PGE_2_ release in a concentration-dependent manner nearly reaching basal PGE_2_ levels in the highest tested concentration of 2.5 µM. LPS also strongly induced TNFα and IL-6 release as well, with no effects by Licochalcone A alone (Figure 2B,C). Licochalcone A significantly reduced LPS induced TNFα-release in the two highest concentrations at about 30% (Figure 2B). IL-6 release was slightly decreased only in the highest concentration of Licochalcone A (Figure 2C). 

### 2.3. Effects of Licochalcone A on the AA/COX/PGE_2_ Pathway

Since we observed a strong reduction of PGE_2_ release, we next focused on the main enzymes of the AA/COX/PGE_2_ pathway. First, we investigated the effects of Licochalcone A on COX-2 and mPGES-1 expression and synthesis using qPCR and Western blot. COX-2 forms PGH_2_ out of arachidonic acid and mPGES-1 supports the reaction of PGH_2_ to PGE_2_; thus, both enzymes exert key roles in the AA/COX/PGE_2_ pathway. LPS strongly induced COX-2 as well as mPGES-1 expression and synthesis in primary rat microglia (Figure 3). Licochalcone A concentration-dependent reversed LPS induced COX-2 synthesis, with a significant reduction of COX-2 protein levels using 1 and 2.5 µM of Licochalcone A (Figure 3A). On mRNA levels, Licochalcone A significantly reduced COX-2 expression in the concentrations of 0.5 and 2.5 µM, but no concentration-dependent effect was observed in the used concentrations (Figure 3B). mPGES-1 synthesis was concentration-dependently inhibited by Licochalcone A with a significant mPGES-1 reduction using 1 and 2.5 µM of Licochalcone A (Figure 3C). In contrast to the reduced mPGES-1 protein levels, mPGES-1 expression was not prevented but was slightly enhanced by Licochalcone A in all tested concentrations (Figure 3D).

Next, we evaluated the effects of Licochalcone A on COX-1 (Figure 4A) and cPLA2 (Figure 4B) synthesis. COX-1 is a constitutively expressed cyclooxygenase, while COX-2 expression is connected to inflammatory stimuli, such as LPS stimulation. COX-1 catalyzes the same reaction as COX-2, while cPLA2 supports the release of AA out of membrane lipids. LPS increases COX-1 synthesis by about 30%, while Licochalcone A concentration-dependent reversed the effects induced by LPS and even decreased COX-1 protein levels below the level of untreated cells (Figure 4A). Only 1 µM and 2.5 µM of Licochalcone A showed a significant reduction of COX-1 levels. cPLA2 synthesis is strongly increased by LPS stimulation (Figure 4B), and Licochalcone A reduced cPLA2 levels starting with the concentration of 1 µM and showing a significant reduction at 2.5 µM Licochalcone A. 

Other than a regulation via the synthesis and expression of the two COX enzymes, PGE_2_ formation might also be regulated via the enzymatic activity of both COX, a major mechanism by which classical non-steroidal anti-inflammatory drugs (NSAIDs) such as aspirin and ibuprofen exert their anti-inflammatory and PGE_2_-inhibiting activities. For that reason, we investigated the effects of Licochalcone A on COX-1 and COX-2 enzymatic activity. As shown in Figure 5A, the availability of AA strongly increased COX-1 activity and thus PGE_2_ formation, while the commercial COX-1 inhibitor SC560 in the concentration of 1 µM potently reduced COX-1 activity by about 60%. Licochalcone A did not show significant effects on COX-1 activity, with slightly enhancing effects in the concentrations of 0.1, 0.5, and 1 µM and slightly suppressing effects in the concentration of 2.5 µM. COX-2 activity is mainly LPS- and AA-dependent. The commercial COX-2 inhibitor diclofenac strongly inhibited COX-2 activity and thus the COX-2-dependent PGE_2_ formation in the concentration of 10 µM (Figure 5B). Licochalcone A decreased COX-2 activity in all tested concentrations up to 50% of the LPS with AA control.

### 2.4. Effects of Licochalcone A on Oxidative Markers

Other than the anti-inflammatory effects of Licochalcone A, we intended to study the effects of this compound on oxidative markers. 8-iso-PGF_2α_ is a sensitive marker for oxidative stress, while NO encounters the activity of nitric oxide synthases and the reactive nitric system (RNS). LPS induced 8-iso-PGF_2α_ release at around 50% (Figure 6A), while NO release was enhanced by about 70% by LPS stimulation (Figure 6B). Neither 8-iso-PGF_2α_ nor NO release was affected by 2.5 µM Licochalcone A without LPS. Licochalcone A showed a concentration-dependent inhibition of 8-iso-PGF_2α_ release, but with a significant reduction only for 2.5 µM of Licochalcone A. LPS-induced NO release was not significantly affected by Licochalcone A treatment.

### 2.5. Molecular Targets

To understand the potentially involved pathways of the observed anti-inflammatory effects of Licochalcone A on the AA/COX-2/PGE_2_ pathway as well as TNFα, IL-6, and 8-iso-PGF_2α_ release, we evaluated the effects of Licochalcone on different signaling pathways, including MAPKs (p38 MAPK, Erk 1/2, Jnk) and the NF-κB pathway. LPS strongly induced the phosphorylation and thus activation of p38 MAPK, Erk 1/2, and JNK, while the synthesis of IκBα (as an inhibitory component of the NF-κB pathway) was strongly suppressed by around 50% (Figure 7). Licochalcone A concentration-dependently decreased phosphorylation of p38 MAPK with a significant reduction using both highest concentrations (Figure 7A). We furthermore observed a concentration-dependent decrease of LPS-induced phosphorylation of Erk 1/2 with significant inhibition in the highest concentration of 2.5 µM of Licochalcone A (Figure 7B). Phospho-JNK showed opposing results with a non-significant concentration-dependent increase of phosphorylation (Figure 7C). For IκBα, Licochalcone A showed an even higher reduction of IκBα levels compared to LPS (Figure 7D). The IκBα reduction was significant in the three highest doses of Licochalcone A compared to the negative control. 

## 3. Discussion

In this study, we investigated the anti-neuroinflammatory and anti-oxidative capacities of Licochalcone A in LPS-stimulated primary rat microglia. We observed a strong reduction of LPS-induced PGE_2_ release after Licochalcone A pretreatment and smaller effects on the release of IL-6 and TNFα. PGE_2_ reduction was accompanied by decreased COX-2 expression and synthesis as well as mPGES-1 synthesis. Furthermore, COX-1 and cPLA2 synthesis as well as COX-2 activity was significantly reduced by Licochalcone A, while COX-1 activity was not significantly affected. On the oxidative side, we observed reduced lipid peroxidation due to the decrease of 8-iso-PGF_2α_ after Licochalcone A treatment, while NO release was not significantly affected. Possible targeted signal pathways responsible for the observed effects are p38 MAPK and Erk 1/2, with decreased phosphorylation of these kinases after Licochalcone A administration.

Anti-neuroinflammatory effects of Licochalcone A were described in a previous study but using the BV2 cell line instead of primary cells. In LPS-stimulated BV2 cells, a murine microglia cell line, PGE_2_ release, as well as COX-2 expression and synthesis were strongly reduced by Licochalcone A pretreatment in a concentration-dependent manner [18]. In contrast to our results, strong effects on IL-6, TNF, and NO release as well as iNOS synthesis/expression and NF-κB signaling by Licochalcone A were demonstrated [18]. These differences might be explained by differences in the used cells. Primary microglia mainly show an innate inflammatory response, and interindividual differences as well as cell-specific responses have less effect on the results since multiple animals are used to gain the cultures. The BV-2 cell line is derived from a single cell clone and is immortalized by introducing two oncogenes after a J2 virus infection, possibly affecting some signaling pathways, even if morphology and function are somehow comparable to primary microglia [42]. Furthermore, the comparison of two different species—mouse for BV-2 cells and rat used for primary microglia cultures in this study—might affect the results, as we have discussed previously due to species-specific differences in receptors, enzymes, and signaling pathways [43]. Additionally, it has been shown that primary microglia underly individual epigenetic priming, modulating inflammatory responses, and result in different reactions of naive microglia to the same stimulus [44], delivering another explanation for the differences observed. Chu et al. showed timepoint-dependent effects of LPS and Licochalcone A in RAW 264.7 cells, a mouse macrophage cell line, possibly explaining differing results as well. After 6 h, they found the strongest induction of TNFα release by LPS and a concentration-dependent decrease of TNFα by Licochalcone A pretreatment [17]. However, after 12 h, Licochalcone A only showed a significant reduction of TNFα in a concentration of 20 µM, and after 24 h LPS stimulation, the timepoint we used in this study, no significant effects of Licochalcone A on TNFα release were found [17], contrasting our results. However, again, two different species, and in this case even different types of cells are compared, possibly explaining differences by functional selectivity and species-dependent effects [43]. Furthermore, both presented studies used 100 times higher doses (1 µg/mL) of LPS for the stimulation of their cell lines and higher total concentrations of Licochalcone A, further hindering a comparison [17,18]. Using LPS-stimulated primary human monocytes, we were only able to confirm the potent effects of Licochalcone A on PGE_2_ release, while IL-6 release was significantly enhanced and TNFα release was not significantly affected by Licochalcone A (Appendix A), thus showing comparable activities in primary human monocytes on those parameters as found in primary rat microglia. Other flavonoids, such as flavone or genistein, showed anti-inflammatory properties by reducing PGE_2_ release and COX-2 synthesis [45], underlining the potential of those classes of derivatives. 

As demonstrated in this study, Licochalcone A inhibited the AA/COX-2/PGE_2_ pathway on different levels of the enzymes involved, leading to a strong reduction of PGE_2_. cPLA2 is activated under inflammatory conditions and initiates the AA/COX-2/PGE_2_ cascade by releasing AA out of the phospholipid membranes of the cells [46]. The observed decrease of cPLA2 synthesis by Licochalcone A therefore most likely reduced the available amount of the substrate AA for final PGE_2_ production. Furthermore, we demonstrated a decreased COX-2 expression and synthesis (of about 50%), and mPGES-1 synthesis was reduced and also contributed to a reduced PGE_2_ synthesis. Taken together, Licochalcone A inhibits key enzymes of the AA/COX-2/PGE_2_ pathway. Even if Licochalcone A does not reduce those enzymes to baseline levels of untreated cells, the synergistic effect of the inhibition on these different cascade levels explains the strong decrease of PGE_2_ release observed. The reduction of COX-2 activity further inhibits the AA/COX-2/PGE_2_ pathway. As a restriction in this study, we found a high COX-2 activity after the addition of AA already, possibly suggesting preactivated microglia. In previous studies, AA administration alone showed only small effects on COX-2 activity measured by released PGE_2_ [43,47]. However, the higher COX-2 activity after the addition of AA might be explained by COX-1 activity as well since the COX-2 assay measures COX-1 and COX-2 activity together. Interestingly, mPGES-1 expression is not affected by Licochalcone A treatment, while its synthesis is reduced. A possible explanation is the use of different time points for our qPCR experiments (4 h) and Western blots (24 h) for mPGES-1. The suppression of mPGES-1 expression might take longer than the inhibition of COX-2 expression and might only be observed later than 4 h. The stimulation of different time points for mPGES-1 expression analysis should be the subject of future studies. COX-1 activity was not significantly affected by Licochalcone A in this current study. However, since we observed a significant decrease in COX-1 synthesis under levels of untreated cells, effects of the gastrointestinal mucosa, as known from other COX-1 inhibitors, cannot be excluded and should be the subject of future research.

MAPK signaling pathways are central in the modulation of inflammatory cell responses, such as cytokine release [16]. In this study, we focused on the three major MAPKs: p38 MAPK, Erk 1/2, and SAPK/JNK. The activation of p38 MAPK leads to the phosphorylation of different targets, such as NF-κB (by acetylation of the p65 subunit [48]), activating transcription factor (ATF)-2, and p53 [16]. Erk 1/2 activates different transcription factors, such as cAMP response element-binding protein (CREB), and especially contributes to cell growth and proliferation but is associated with inflammation as well [49]. Both, p38 MAPK and Erk 1/2 are involved in cPLA2 activation in smooth muscles after muscarinic stimulation [50]. cPLA2 amplified COX-2 expression and oxidative stress leading to higher PGE_2_ levels and neuron loss after middle cerebral artery occlusion in wildtype mice when compared to cPLA2 knockout mice [26]. Phosphorylation of SAPK/JNK enhances pro-inflammatory cytokine release, such as TNFα, and induces apoptosis [16]. Furthermore, activation of p38 MAPK increases COX-2 expression as well as COX-2 mRNA stability, while NF-κB only enhances COX-2 expression via another pathway than p38 MAPK [35]. As we have shown before, protein kinase C (PKC) is another important kinase in the regulation of COX-2 expression, and inhibition of PKC strongly reduced LPS-induced COX-2 expression in primary rat microglia [51]. Furthermore, phosphoinositide 3 kinase (PI3K) reduced mPGES-1 levels while enhancing COX-2 synthesis in LPS-activated primary rat microglia, suggesting an at least partially independent regulation of COX-2 and mPGES-1 [28]. In the current study, we just investigated the effects of Licochalcone A on IκBα, a molecule inhibiting the nuclear translocation of NF-κB and therefore preventing its activation [52]. Compared to untreated cells, Licochalcone A further reduced the LPS-induced decrease of IκBα, suggesting a slight enhancement of LPS-induced NF-κB activation. However, the effects of Licochalcone A on p38 MAPK and Erk 1/2 were revealed to be more pronounced than the effect on IκBα, with a markable decrease of IκBα synthesis after LPS stimulation already. Future studies should assess the effects of Licochalcone A on NF-κB in primary microglia, especially since inhibitory effects on phosphorylation of NF-κB by Licochalcone A were demonstrated in LPS-stimulated BV-2 cells [18]. mPGES-1 is regulated by the mentioned MAP-kinases as well, with SAPK/Jnk enhancing and p38 MAPK and Erk 1/2 negatively modulating mPGES-1 expression [53]. mPGES-1 might, however, not be regulated on expression level but rather posttranscriptional since the regulation of mPGES-1 mRNA stability by JNK has been described [54]. Taken together, the observed effects of Licochalcone A on LPS-induced p38 MAPK, Erk 1/2, and SAPK/JNK phosphorylation possibly explain the shown effects on the AA/COX-2/PGE_2_ cascade. The effects of Licochalcone A on the phosphorylation of p38 MAPK and Erk 1/2 were reproduced in LPS-induced acute lung injury mice models [17], while Huang et al. only showed a decrease in phosphorylation for Erk 1/2 and not for p38 MAPK in LPS-stimulated BV-2 cells [18]. 

PGE_2_ itself further enhances COX-2 and mPGES-1 expression via EP2 receptors; thus, a reduction of PGE_2_ should be accompanied by reduced COX-2 and mPGES-1 expression and synthesis [55] as well as reduced cytokine release [25] via different signaling pathways. Furthermore, the possible involvement of 8-iso-PGF_2α_ as a signaling molecule of the AA/COX-2 pathway should be considered and further investigated in future studies, as we have discussed previously [56].

The observed effect of Licochalcone A on 8-iso-PGF_2α_ release supports the hypothesis that Licochalcone A possesses anti-oxidative capacities. 8-iso-PGF_2α_ is known as a sensitive marker for oxidative stress and is produced by the lipid peroxidation of AA and in much smaller amounts by cyclooxygenases [36]. In contrast to the study by Huang et al., we were not able to reproduce the effects of Licochalcone A on reactive nitrogen species. In the cited study, a decrease of nitrite release using Griess reagent as well as iNOS expression and synthesis was observed in LPS-stimulated BV-2 cells [18]. In our study, we could not confirm the effect of Licochalcone A on NO release in LPS-activated primary microglia. Again, species-specific and cell-dependent effects might be relevant, as discussed above. Chu et al. reported reduced myeloperoxidase activity, a key enzyme of oxidative damage caused by the respiratory burst of activated neutrophils, as well as a higher activity of superoxide dismutase 1 (SOD1), an important anti-oxidative enzyme, in LPS-induced murine models of acute lung injury [17]. These results further support the anti-oxidative properties of Licochalcone A. Oxidative stress is associated with different disorders and can contribute to neurodegeneration and therefore neurological and neuropsychiatric disorders [8,40,57]. This study mainly focuses on the inhibitory effects of Licochalcone A on inflammatory parameters. However, the reduction of 8-iso-PGF_2α_ is promising and the effects of Licochalcone A on oxidative stress should be further evaluated in future research using primary cell cultures and disease models. 

Inflammation and oxidative stress both lead to neurodegeneration, promoting each other [38], and are closely connected to neurological and neuropsychiatric disorders [4,40,58]. However, treatment of disorders such as AD, PD, and MS still remains symptomatic and can be associated with severe side effects. Therefore, new compounds with anti-inflammatory capacities might offer new opportunities in the treatment of chronic inflammation in the pathophysiology of these diseases. Adding anti-oxidative effects to the already used anti-inflammatory mechanisms might boost the success of the treatment in those disorders even more. The effects of Licochalcone A on LPS-induced murine Parkinson model have already been shown to reduce motoric dysfunction [18]. Intravenous administration of Licochalcone A improved the cognitive ability and spatial memory of middle-aged mice in the Morrison water maze test [19]. Therefore, further research in animal models might reveal other disorders and diseases that might benefit from the anti-inflammatory and anti-oxidative effects of Licochalcone A. 

## 4. Materials and Methods

### 4.1. Compound Preparation

Purified Licochalcone A was purchased from Sigma-Aldrich GmbH (Taufkirchen, Germany) and suspended in DMSO prior to experiments. According to the manufacturer, the purity is ≥96%, as determined by HPLC. Lipopolysaccharide (LPS) from Salmonella typhimurium was purchased (Sigma-Aldrich GmbH, Taufkirchen, Germany), resuspended in sterile phosphate-buffered saline (PBS) as 5 mg/mL stock and used at a final concentration of 10 ng/mL in the experiments.

### 4.2. Primary Microglia Cultures

#### 4.2.1. Ethics Statement

Animals were obtained from the Center for Experimental Models and Transgenic Services-Freiburg (CEMT-FR). All the experiments were approved and conducted according to the guidelines of the ethics committee of the University of Freiburg Medical School under protocol No. X-13/06A, and the study was carefully planned to minimize the number of animals used and their suffering.

#### 4.2.2. Primary Rat Microglia Cultures

Primary microglia cultures were prepared from cerebral cortices of one-day neonatal Wistar rats as previously described [59,60,61]. In brief, brains were carefully removed and cerebral cortices were collected and freed from meninges. Further, forebrains were minced, gently dissociated by repeated pipetting in Dulbecco’s modified Eagle’s medium (DMEM), and filtered by passing through a 70 µm nylon cell strainer (BD biosciences, Heidelberg, Germany). Cells were then collected by centrifugation (1000× *g*, 10 min) and resuspended in DMEM medium containing 10% fetal calf serum (FCS) (GE Healthcare, Freiburg, Germany) plus 1% penicillin and streptomycin (40 U/mL and 40 µg/mL, respectively; Sigma-Aldrich GmbH, Taufkirchen, Germany). Cells were cultured on 10 cm cell culture dishes (Falcon, Heidelberg, Germany) with a density of 5 × 10^5^ cells/mL in 5% CO_2_ at 37 °C. After 12–14 days in vitro, floating microglia were harvested from mixed glia (astrocyte-microglia) cultures and re-seeded into cell culture plates (Falcon, Heidelberg, Germany) at a density of 2 × 10^5^ cells/well. On the next day, medium was removed to get rid of non-adherent cells, a fresh medium was added, and, after 1 h, cells were used for experiments.

### 4.3. Cell Viability Assay

Viability of primary rat microglia after treatment with Licochalcone A was measured by the CellTiter-Glo^®^ Luminescent Cell Viability Assay (Promega Corporation, Madison, WI, USA). The assay is used to determine the number of metabolically active and viable cells in cell culture based on quantification of ATP in the cells. Cells (2 × 10^5^/mL) were cultured for 24 h and then incubated with LPS (10 ng/mL) or Licochalcone A (0.1, 0.5, 1, and 2.5 μM) for 24 h. The concentration of ATP was measured after adding 100 µL of reconstituted substrate and additional incubation time of 10 min. Thereafter, luminescence was measured using GloMax^®^ Luminometer (Promega Corporation, Madison, Wisconsin, USA).

### 4.4. Determination of PGE_2,_ 8-iso-PGF_2α_, IL-6, and TNFα Production from LPS Activated Primary Microglia by ELISA

Cultured primary rat microglia were stimulated with LPS (10 ng/mL) alone or after pretreatment with Licochalcone A (0.1, 0.5, 1, and 2.5 µM) for 24 h. Supernatants were collected and centrifuged at 5000× *g* for 5 min at 4 °C. PGE_2_ and 8-iso-PGF_2α_ production was assessed in cell supernatants with commercially available enzyme immunoassay (EIA) kits (Cayman Chemicals, Ann Arbor, MI, USA), respectively, followed by measurement at 450 nm according to the manufacturer’s protocol. Release of IL-6 and TNFα in cell supernatants was measured with a commercially available ELISA kit (eBioscience Inc., Frankfurt, Germany) following the manufacturer’s assay protocol and measurements at a wavelength of 450 nm. Briefly, for the IL-6 and TNFα ELISA, 96-well plates (Thermo Fisher Scientific, Bonn, Germany) were coated with the respective antibodies. On the next day, plates were blocked and washed, then standards as well as supernatants were added to the wells in previously determined dilutions. Detection antibody, Streptavidin-HRP, and stop solution were added in the following steps followed by the measurement. For the PGE_2_ and 8-iso-PGF_2α_ EIA, standards or supernatants in empirical dilutions were pipetted onto the IgG-coated plates, and AChE tracer and PGE_2_monoclonal antibodies were added to all wells. On the next day, plates were washed and Ellman’s reagent added before measurement. Experiments were performed at least three times. Data were normalized to LPS control and presented as percentage change.

### 4.5. Western Blot

Rat primary microglia were stimulated with LPS (10 ng/mL) alone or in presence of Licochalcone A (0.1, 0.5, 1, and 2.5 µM) for 30 min (signaling) or 24 h (COX-2, mPGES-1, COX-1, cPLA2). After stimulation, cells were washed with cold phosphate-buffered saline (PBS) and lysed with lysis buffer (42 mM Tris-HCl, 1.3% sodium dodecyl sulfate (SDS), 6.5% glycerin, 100 μM sodium orthovanadate, and 2% phosphatase and protease inhibitors). Protein concentrations of the samples were measured using the bicinchoninic acid protein (BCA) assay kit (Thermo Fisher Scientific, Bonn, Germany) according to the manufacturer’s instructions. For Western blots, 15–20 µg of total protein from each sample was subjected to SDS-PAGE under reducing conditions. Afterward, proteins were transferred onto polyvinylidene fluoride (PVDF) membranes (Merck Millipore, Darmstadt, Germany). After blocking with 5% milk solution (Bio-Rad Laboratories GmbH, Feldkirchen, Germany) in Tris-buffered saline (TBS) containing 0.1% Tween 20 (TBS-T), membranes were incubated with primary antibodies. Primary antibodies used were goat anti-COX-2 (sc-1745, Santa Cruz Biotechnology Inc., Heidelberg, Germany; 1:500), rabbit anti-mPGES-1 (No. 160140, Cayman Chemical Co., Ann Arbor, MI, USA; 1:6000), anti-p42/44 MAPK (Erk 1/2) (9102S, Cell Signaling Technology Inc., Danvers, MA, USA; 1:1000), anti-p38 MAPK (9212S, Cell Signaling Technology Inc., Danvers, MA, USA; 1:1000), anti-SAPK/Jnk (9252S, Cell Signaling Technology Inc., Danvers, MA, USA; 1:1000), anti-phospho p38 MAPK (9211S, Cell Signaling Technology Inc., Danvers, MA, USA; 1:1000), anti-phospho SAPK/Jnk (9251S, Cell Signaling Technology Inc., Danvers, MA, USA; 1:1000), anti-phospho p42/44 MAPK (Erk 1/2) (9101S, Cell Signaling Technology Inc., Danvers, USA; 1:1000), anti-phospho IkB (sc-8404, Santa Cruz Biotechnology Inc., Heidelberg, Germany; 1:500), and rabbit anti-β-actin (SAB5600204, Sigma-Aldrich GmbH, Taufkirchen, Germany; 1:5000). Primary antibodies were diluted in TBS-T and 5% BSA. Membranes were incubated with the primary antibody overnight at 4 °C, followed by incubation with the secondary antibodies. After extensive washing (three times for 15 min each in TBS containing 0.1% Tween 20), proteins were detected with horseradish peroxidase (HRP)-coupled anti-goat IgG (Santa Cruz Biotechnology Inc., Heidelberg, Germany) and anti-rabbit IgG (RD systems, Wiesbaden, Germany) using enhanced chemiluminescence (ECL) reagents (Biozym, Hessisch Oldendorf, Germany) and a LICOR Odyssey Imager (LI-COR Corporate, Lincoln, NE, USA). Equal protein loading and transfer were assessed by subjection of each sample to a Western blot for actin or the corresponding total protein for signaling. All Western blot experiments were carried out at least three times. Densitometric analysis was performed using ImageJ software (NIH, Bethesda, MD, USA). 

### 4.6. Real-Time PCR

Quantitative PCR (qPCR) was performed to determine the transcriptional regulation of COX-2 and mPGES-1 by Licochalcone A in activated microglia. RNA preparation was performed by using RNAspin mini RNA isolation kit (GE Healthcare, Freiburg, Germany), and for cDNA synthesis 1 µg of total RNA was reverse transcribed using M-MLV reverse transcriptase (Promega, Mannheim, Germany) and random hexamers (Biomers.net, Ulm, Germany). The synthesized cDNA was the template for the qPCR amplification that was carried out by the CFX96 real-time PCR detection system (Bio-Rad Laboratories GmbH, Feldkirchen, Germany) using iQ^TM^ SYBR^TM^ Green supermix (Bio-Rad Laboratories GmbH, Munich, Germany). 

Specific primers (Biomers.net, Ulm, Germany) were designed by using Universal Probe Library Assay Design Center (Roche Diagnostics, Mannheim, Germany). Reaction conditions were 3 min at 95 °C, followed by 40 cycles of 15 s at 95 °C, 30 s at 50 °C, and 45 s at 72 °C, and every cycle was followed by plate read. After that, 1 min at 95 °C, 1 min at 55°C, followed by melt curve conditions of 65 °C, 95 °C with increment of 0.5 °C for 5 s, followed by final plate read. GAPDH served as an internal control for sample normalization and the comparative cycle threshold Ct method was used for data quantification as described previously [62]. The following primer sequences were used in the present study. COX-2: Forward 5′-CTACACCAGGGCCCTTCC-3′; Reverse 5′-TCCAGAACTTCTTTTGAATCAGG-3′, mPGES-1: Forward 5′-GCACACTGCTGGTCATCAAG-3′; Reverse 5′-ACGTTTCAACGCGTCCTC-3′, GAPDH: Forward 5′-TGGGAAGCTGGTCATCAAC-3′, Reverse 5′-GCATCACCCCATTTGATGTT-3′.

### 4.7. Cyclooxygenase Activity Assay

COX enzymatic activity was measured using the arachidonic acid assay, as described previously [63]. For COX-1 activity, cells were plated in 24-well plates, and after 24 h, medium was removed and replaced with serum-free medium. Licochalcone A (0.1, 0.5, 1, and 2.5 μM) or the selective inhibitor of COX-1 SC560 [(1 μM); Sigma-Aldrich GmbH, Taufkirchen, Germany] were added and left for 15 min. Then, arachidonic acid (15 μM; Sigma-Aldrich GmbH, Taufkirchen, Germany) was applied for another 15 min. Finally, supernatants were collected and used for the determination of PGE_2_ as described above. 

For COX-2 enzymatic activity, the assay was conducted as for COX-1, but with pre-incubation with LPS (10 ng/mL) for 24 h to induce COX-2 synthesis and using diclofenac sodium [(10 μM); Sigma-Aldrich GmbH, Taufkirchen, Germany] as commercial COX-2 preferential inhibitor.

### 4.8. NO Release Assay

Primary rat microglia were pretreated with Licochalcone A (0.1, 0.5, 1, and 2.5 µM) for 30 min. Then, cells were stimulated with 10 ng/mL LPS and incubated for 24 h. Supernatants were harvested, and NO levels measured using the commercially available Griess Reagent System (Promega Corporation, Madison, WI, USA) following the manufacturer’s protocol. In brief, a NO standard curve was prepared, and standards as well as supernatants of the samples were transferred to a 96-well microplate. Sulfanilamide solution was added to all wells, and the plate was incubated for 10 min in the dark. Then, N-1-naphthylethylenediamine dihydrochloride (NED) solution was added to all wells and the plate was incubated for 10 min again. The plate was read at 530 nm using a colorimetric microplate reader (MRXe Microplate reader, Dynex Technologies, Denkerdorf, Germany).

### 4.9. Statistical Analysis

Raw values were converted to percentage and LPS (10 ng/mL), or the appropriate positive control, such as untreated cells for ATP-assay, were considered as 100%. Values of all experiments are presented as mean ± SEM of at least three independent experiments. Values were compared using one-way ANOVA with Dunett’s post hoc test (Prism 8 software, GraphPad Software Inc., San Diego, USA). The level of significance was set at * *p* < 0.05, ** *p* < 0.01, *** *p* < 0.001, and **** *p* < 0.0001.

## 5. Conclusions

Neuroinflammation and oxidative stress are both conditions leading to neurodegeneration and, in consequence, to neurological as well as neuropsychiatric disorders. However, there is still a lack of highly effective treatment. In this study, we show that the flavonoid Licochalcone A exerts anti-neuroinflammatory and anti-oxidative effects in primary rat microglia mainly dependent on the arachidonic acid/COX-2/PGE_2_ pathway. Especially, p38 MAPK and Erk 1/2 signaling seem to be relevant for the observed effects. Therefore, future research using animal disease models should further investigate the effects of Licochalcone A on the pathophysiology of different neuropsychiatric disorders.

## Figures and Tables

**Figure 1 molecules-28-01927-f001:**
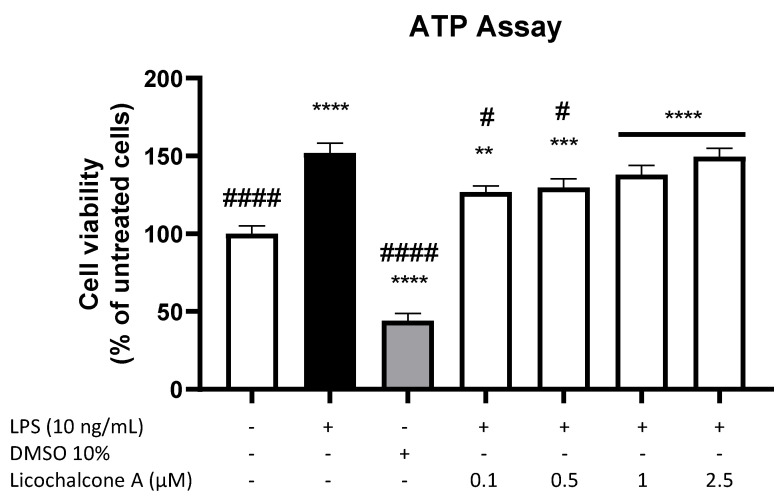
Effects of Licochalcone A on cell viability of primary rat microglial cells. Cells were stimulated as described under materials and methods and, after 24 h, ATP was quantified using the CellTiter-Glo^®^ Luminescent Cell Viability Assay. Values are presented as the mean ± SEM of at least three independent experiments. Statistical analysis was performed using one-way ANOVA with Dunnett’s post hoc tests with # *p* < 0.05, ** *p* < 0.01, *** *p* < 0.001, and ****/#### *p* < 0.0001 compared to untreated cells (stars *) and LPS-stimulated cells (hashtags #).

**Figure 2 molecules-28-01927-f002:**
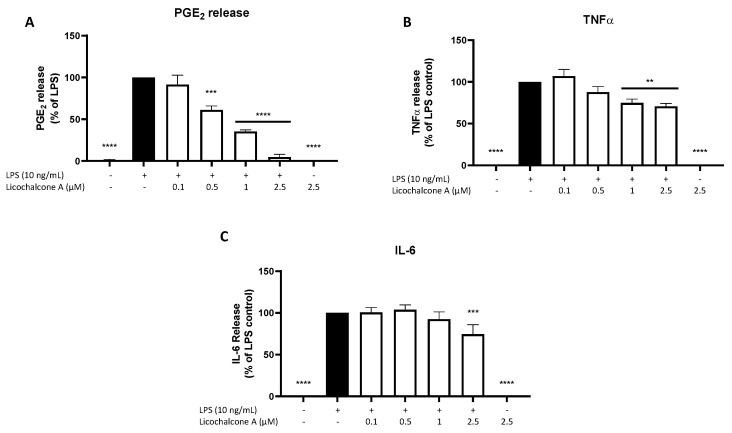
Effects of Licochalcone A on PGE_2_ (**A**), TNFα (**B**), and IL-6 (**C**) release in LPS-stimulated primary rat microglial cells. Cells were stimulated as described under materials and methods and supernatants were collected for EIA/ELISA measurements after 24 h. Values are presented as the mean ± SEM of at least three independent experiments. Statistical analysis was performed using one-way ANOVA with Dunnett’s post hoc tests with ** *p* < 0.01, *** *p* < 0.001, and **** *p* < 0.0001 compared to LPS-stimulated cells.

**Figure 3 molecules-28-01927-f003:**
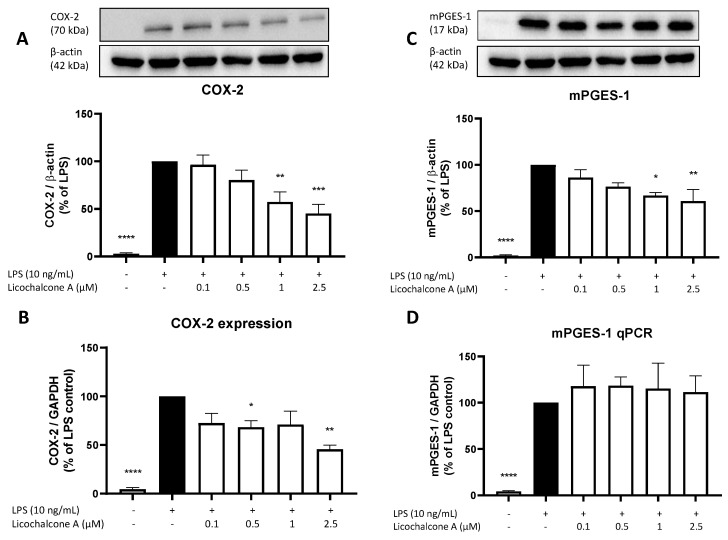
Effects of Licochalcone A on COX-2 (**A**,**B**) and mPGES-1 (**C**,**D**) synthesis (**A**,**C**) and expression (**B**,**D**) in LPS-stimulated primary rat microglial cells. Cells were stimulated and Western Blot/qPCR performed as described under materials and methods. Values are presented as the mean ± SEM of at least three independent experiments. Statistical analysis was performed using one-way ANOVA with Dunnett’s post hoc tests with * *p* < 0.05, ** *p* < 0.01, *** *p* < 0.001, and **** *p* < 0.0001 compared to LPS stimulated cells.

**Figure 4 molecules-28-01927-f004:**
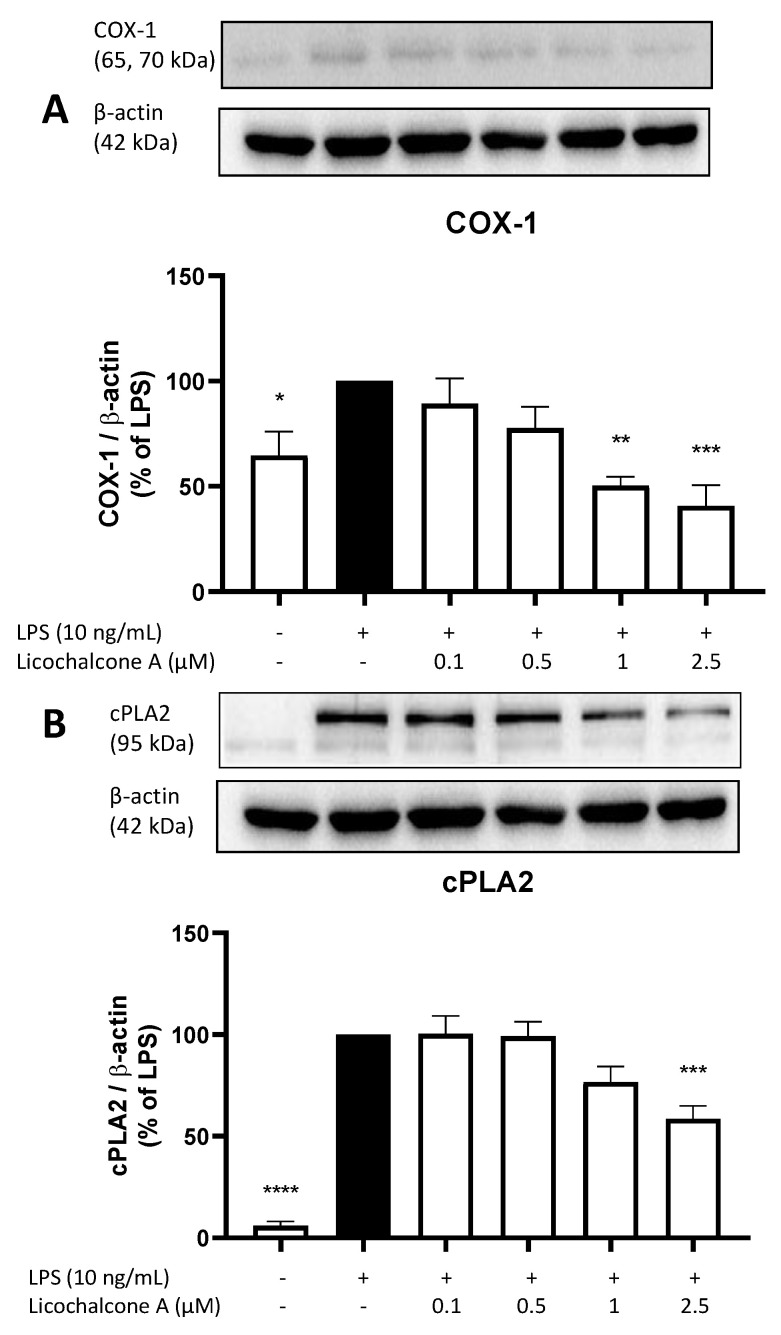
Effects of Licochalcone A on COX-1 (**A**) and cPLA2 (**B**) synthesis in LPS-stimulated primary rat microglial cells. Cells were stimulated and Western Blot performed as described under materials and methods. Values are presented as the mean ± SEM of at least three independent experiments. Statistical analysis was performed using one-way ANOVA with Dunnett’s post hoc tests with * *p* < 0.05, ** *p* < 0.01, *** *p* < 0.001, and **** *p* < 0.0001 compared to LPS stimulated cells.

**Figure 5 molecules-28-01927-f005:**
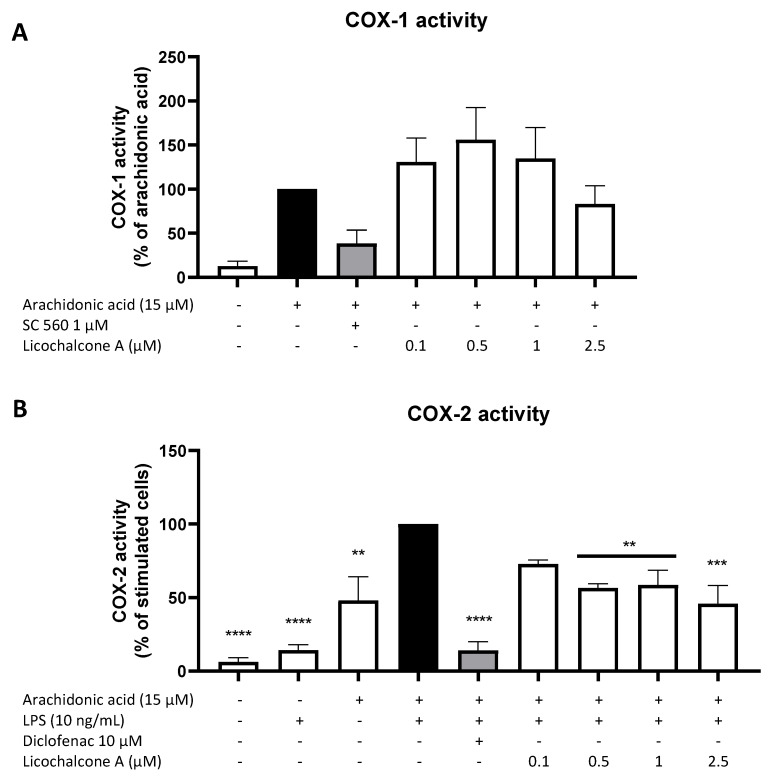
Effects of Licochalcone A on COX-1 (**A**) and COX-2 (**B**) activity in primary rat microglial cells. Cells were stimulated and COX-1/COX-2 activity assays performed as described under materials and methods. Values are presented as the mean ± SEM of at least three independent experiments. Statistical analysis was performed using one-way ANOVA with Dunnett’s post hoc tests with ** *p* < 0.01, *** *p* < 0.001, and **** *p* < 0.0001 compared to AA (**A**) or AA + LPS (**B**) stimulated cells.

**Figure 6 molecules-28-01927-f006:**
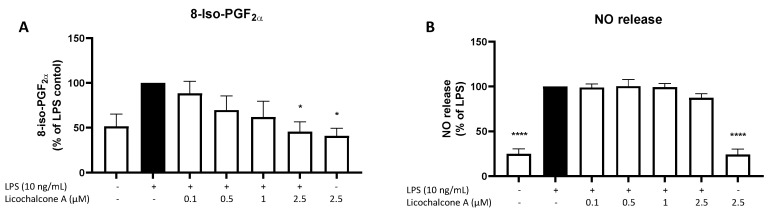
Effects of Licochalcone A on 8-iso-PGF_2α_ (**A**) and NO (**B**) release in LPS-stimulated primary rat microglial cells. Cells were stimulated and 8-iso-PGF_2α_ EIA or the NO Griess reagent assay performed as described under materials and methods. Values are presented as the mean ± SEM of at least three independent experiments. Statistical analysis was performed using one-way ANOVA with Dunnett’s post hoc tests with * *p* < 0.05, and **** *p* < 0.0001 compared to LPS-stimulated cells.

**Figure 7 molecules-28-01927-f007:**
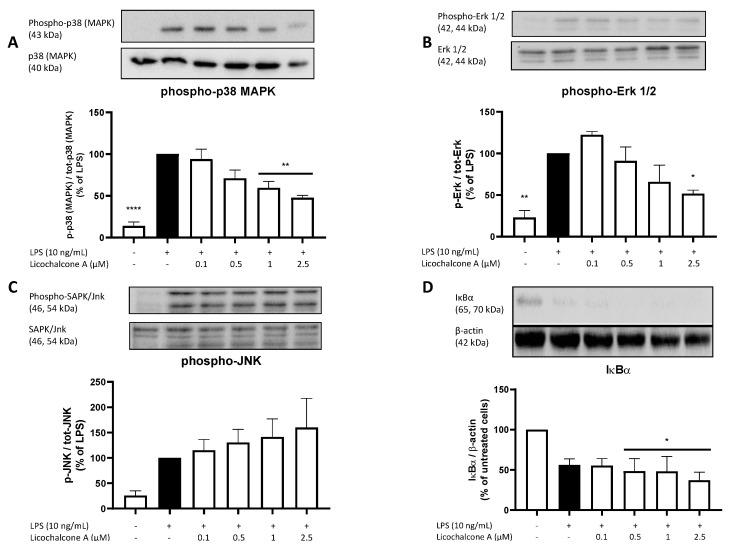
Effects of Licochalcone A on phospho-p38 MAPK (**A**), phospho-Erk 1/2 (**B**), phospho-JNK (**C**), and IκBα (**D**) in LPS-stimulated primary rat microglial cells. Cells were stimulated and Western blots performed as described under materials and methods. Values are presented as the mean ± SEM of at least three independent experiments. Statistical analysis was performed using one-way ANOVA with Dunnett’s post hoc tests with * *p* < 0.05, ** *p* < 0.01, and **** *p* < 0.0001 compared to LPS stimulated (**A**–**C**) or untreated (**D**) cells.

## Data Availability

The data presented in this manuscript are available from the corresponding author upon request.

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
