# Peer review of "Licochalcone A Inhibits Prostaglandin E2 by Targeting the MAPK Pathway in LPS Activated Primary Microglia"

_molecules, 2023, doi:10.3390/molecules28041927_

Round 1

Reviewer 1 Report

The study aims to investigate the inflammatory and anti-oxidative effects of Licochalcone A in LPS activated primary rat microglia. The introduction of the manuscript is detailed and comprehensive, although only 6 out of 41 references are from the last 3 years.

The results and discussion section is well structured covering the effect of Licochalcon A on variety of experimental settings. However in the Material and Methods section better description of the methods used is recommended. In L444 – more details about the manufacturer protocol should be given. For description of Cyclooxygenase activity assay, in L 506, reference 63 is cited, which is not included in the list of References.

Author Response

Dear Reviewer 1, many thanks for taking the time and reviewing our manuscript. We have added some more details in L444 regarding the manufacturers’ protocols. Furthermore, we have added the missing reference 63 for the COX activity assay. Thank you for noticing this missing reference!

Reviewer 2 Report

1. In the figure showing the results of all data, it is necessary to indicate the significance of the normal group, the LPS-treated group, and the drug-treated group.

2. In the results of Figure 5A, it is desirable to express the significance that AA increased the activity of cox-1 compared to the normal group.

3. Many studies have already been reported that confirmed the efficacy of inflammation by inducing Licochalcone A with LPS. The authors believe that it is necessary to assert the difference between the subject of the manuscript and previous studies in more detail.

Author Response

  1. In the figure showing the results of all data, it is necessary to indicate the significance of the normal group, the LPS-treated group, and the drug-treated group.

Answer: Dear Reviewer 2, many thanks for taking the time and reviewing our manuscript. We added the significance of LPS against negative control (untreated cells) and Licochalcone A without LPS in all figures additionally. For the ATP-assay we included significances of LPS against all other groups beside the comparison against the negative control.

  1. In the results of Figure 5A, it is desirable to express the significance that AA increased the activity of cox-1 compared to the normal group.

Answer: In the Figure 5A, AA does not significantly enhance the COX-1 activity compared to the negative control. Therefore, no significances can be shown for any comparison in Figure 5A.

  1. Many studies have already been reported that confirmed the efficacy of inflammation by inducing Licochalcone A with LPS. The authors believe that it is necessary to assert the difference between the subject of the manuscript and previous studies in more detail.

Answer: As we state in our introduction, we believe that chronic low-grade inflammation might be associated with different neuropsychiatric disorders. Previous studies focusing on effects of Licochalcone A used higher doses of LPS compared to this study, possibly activating different pathways and alternated immune responses of the cells. Furthermore, we used primary microglial cells, whereas other studies used cell lines, that might show different inflammatory responses due to their mutation events (line 95-97; 261-266).